# Early Light Chains Removal and Albumin Levels with a Double Filter-Based Extracorporeal Treatment for Acute Myeloma Kidney

**DOI:** 10.3390/toxins14060391

**Published:** 2022-06-05

**Authors:** Gabriele Donati, Fulvia Zappulo, Elisa Maietti, Anna Scrivo, Lorenzo Gasperoni, Elena Zamagni, Paola Tacchetti, Lucia Pantani, Olga Baraldi, Giorgia Comai, Maria Cappuccilli, Michele Cavo, Gaetano La Manna

**Affiliations:** 1Nephrology, Dialysis and Renal Transplantation Unit, Azienda Ospedaliero-Universitaria di Modena, Surgical, Medical, Dental and Morphological Sciences Department (CHIMOMO), University of Modena and Reggio Emilia, 41124 Modena, Italy; gabriele.donati@unimore.it; 2Nephrology Dialysis and Renal Transplantation Unit, IRCCS Azienda Ospedaliero-Universitaria di Bologna, Department of Experimental Diagnostic and Specialty Medicine (DIMES), Alma Mater Studiorum University of Bologna, 40138 Bologna, Italy; fulvia.zappulo@studio.unibo.it (F.Z.); anna.scrivo@aosp.bo.it (A.S.); lorenzo.gasperoni3@gmail.com (L.G.); olga.baraldi@aosp.bo.it (O.B.); giorgia.comai@aosp.bo.it (G.C.); maria.cappuccilli@unibo.it (M.C.); 3Department of Biomedical and Neuromotor Sciences-DIBINEM, University of Bologna, 40138 Bologna, Italy; elisa.maietti2@unibo.it; 4Hematology and Oncology Unit “Lorenzo & Ariosto Seràgnoli”, IRCCS Azienda Ospedaliero-Universitaria di Bologna, Alma Mater Studiorum University of Bologna, 40138 Bologna, Italy; e.zamagni@unibo.it (E.Z.); paola.tacchetti@unibo.it (P.T.); lucia.pantani2@unibo.it (L.P.); michele.cavo@unibo.it (M.C.)

**Keywords:** multiple myeloma, acute kidney injury, double dialysis filter, PEPA, PMMA, free light chains

## Abstract

Renal impairment in Multiple Myeloma (MM) represents one of the most important factors that influences patient survival. In fact, before the introduction of modern chemotherapy, less than 25% of patients with acute kidney injury (AKI) and MM who required dialysis recovered sufficient renal function to become independent from dialysis, with a median overall survival of less than 1 year. There are many other factors involved in determining patient survival. In this study we aimed to investigate the role of double filter-based extracorporeal treatment for removal of serum free light chains (sFLC) in acute myeloma kidney (AKI for MM) and to evaluate patient overall survival. All patients received Bortezomib-based chemotherapy and extracorporeal treatment for sFLC removal. For each session 2 dialyzers of the same kind were used. The dialytic dose was not related to the degree of renal function but to the removal of sFLC. The factors that have been found to be significantly associated with lower mortality were reduction of sFLC at day 12 and day 30, >50% reduction of sFLC at day 30, number of sessions and independence from dialysis. Among baseline characteristics, albumin level was statistically associated with the patients’ outcome. Our analysis highlights the importance of the early treatment for removal of sFLC in AKI for MM. These results indicate that the early removal of sFLC can improve patient’s outcome.

## 1. Introduction

Multiple myeloma (MM) is the second most frequent hematological malignancy [1]. Renal injury in MM can derive from different mechanism of injury and could be clinically manifest with Acute Kidney Injury (AKI), proteinuria or both [2]. 50% of new diagnosed MM patients show serum creatinine >1.3 mg/dL, 12–20% show serum creatinine >2.0 mg/dL and about 10% are affected by acute renal failure requiring dialysis [3]. Haynes et al. demonstrated that the recovery of renal function is of paramount importance for the survival of patients with MM with acute renal failure with or without dialysis need [4]. The same authors highlighted that albumin values >35 g/L and the eligibility to chemotherapy had also a pivotal role in patients’ survival [4]. Nonetheless renal function recovery depends mainly on serum free light chains (sFLC) removal. Leung et al. firstly assessed that those patients who experienced renal function recovery obtained a sFLC reduction >50% combining chemotherapy and plasmapheresis [5]. Between patients with renal function recovery, kidney biopsy revealed that the main cause of acute renal injury (AKI) was myeloma cast nephropathy (MCN). Therefore, the efforts to provide the extracorporeal removal of sFLC combined with chemotherapy were addressed to those kinds of patients. Nonetheless, Gupta et al. considering a pooled analysis of data from 3 randomized and controlled trial and 2 retrospective studies did not find a clear benefit of plasmapheresis for AKI from MCN [6]. In the following years, Hutchison et al. proposed high cut off dialysis (HCO HD) as a new extracorporeal technique for sFLC removal. He proved that HCO HD was superior to plasmapheresis in providing recovery from AKI, but with high rate of albumin transfusion due to the high albumin leakage from the HCO membrane [7]. When HCO HD, combined to chemotherapy, provides a sFLC reduction >50% at day 12 and at day 21, the renal recovery significantly improves [7]. These assumptions were undermined by two randomized and controlled trials. The first was the MYRE trial that did not find any significant survival improvement after 12 months for those patients who underwent HCO HD and chemotherapy in comparison to those who underwent conventional HD and chemotherapy [8]. Two years later Huchison et al. published the results of the European Trial of Free Light Chain Removal by Extended Haemodialysis in Cast Nephropathy (EuLITE): patients who underwent HCO HD with chemotherapy did not show any improvement of the renal function after 24 months and showed a high mortality rate in comparison to those patients who underwent high flux dialysis and chemotherapy [9].

The aim of the present observational study is to assess patient’s survival and compare 3 kinds of high flux dialyzers with adsorbing properties, combined to chemotherapy, for sFLC removal in patients affected by AKI from newly diagnosed MM.

## 2. Results

### 2.1. Study Population

Twenty-three patients were enrolled in this study. All patients were treated with bortezomib-based chemotherapy and extracorporeal treatment for sFLC removal. Patient’s characteristics have been described in Table 1. In this case, 16 patients were men while seven were women. Median age was 67 years old (min 41, max 85 years). Median eGFR was 13.7 mL/min (min 2.8, max 57 mL/min). Glomerular Filtration Rate was estimated through the equation CKD-EPI (Chronic Kidney Disease Epidemiology Collaboration). Here, 18 patients presented urine output lower than 500 mL/day, 5 patients had urine output between 500 and 1000 mL/day. The most common monoclonal protein was the sub-type kappa in 16 patients (69.6%) while 7 patients (30.4%) had lambda light chains. Median level of pathological sFLC was 4698.1 mg/L (range 645–22,286). In particular, we found a median level of kappa light chain of 3,475.6 mg/L (range 1000.1–15,768) and a median level of lambda light chain of 11,279 (range 645–22,286) mg/L.

Median albumin was 3.6 mg/dL (min 2.3–max 4.7 mg/dL). In this case, 17 patients (73.9%) presented a serious renal injury with eGFR lower than 15 mL/min (grade 5 classification NKF-KDOQI), 18 patients (78.3%) presented AKI class III according to AKIN criteria at presentation with the need of urgent dialytic treatment, 5 patients (21.7%) presented AKI class II with no need for renal replacement therapy, however extracorporeal treatment was performed to remove sFLC before the result of kidney biopsy. Nonetheless patients with AKI class II were equally distributed among the 3 groups: 1/5 patients with AKI class II were in PMMA group, 2/5 were in FDX group, 2/5 patients were in FDY group. In 18 patients (78.3%) kidney biopsy was performed. MCN was found in 14 patients, in 3 cases monoclonal immunoglobulin deposition disease (MIDD) was found and in 1 case light chain deposition disease (LCDD) was found. In 5 patients kidney biopsy was not performed because high bleeding risk due to platelets count <30,000/mmc or to clinical condition of the patient. For these patients the presumptive diagnosis of MCN was based, according to previous study, on the following criteria: (a) new diagnosis of MM; (b) sFLC above 500 mg/L; (c) albumin to creatinine ratio <30%; (d) persistence of AKI after correction of precipitant factors. (hypercalcemia, nephrotoxic agents, dehydration) [10].

9 patients (23%) underwent chemotherapy with bortezomib, dexamethasone and thalidomide; 13 patients (52%) with bortezomib and dexamethasone. 1 patient (4%) has been treated with bortezomib, cyclophosphamide and dexamethasone. 3 patients underwent a second-line chemotherapy because of non-response to the first line therapy: in 2 patients, thalidomide was added on the therapy while one patient was treated with daratumumab.

#### 2.1.1. Dialyzers

The 3 filters used were high adsorbitive dialyzers. Seven patients underwent extracorporeal treatment through polymethylmetacrilate (PMMA)-Filtryzer BK-F using bicarbonate hemodialysis; 7 patients have been treated with polyester polymer alloy (PEPA) dialyzer FDX-210 GW and 9 patients with PEPA FDY-210 GW both with on-line hemodiafiltration (ol-HDF). All the treatments were well tolerated. Statistical analysis showed that there were no significant differences in the principal variables at baseline in the 3 groups (Table 2). Furthermore, there was no significant difference between the 3 groups in removal of sFLC: RR median per session was 50.4% in patient treated with Filtryzer BK-F, 51.2% with FDX-210 GW; 49.7% with FDY-210 GW; *p* = 0.698). Similarly, the degree of reduction of sFLC at day 12 and day 30 from the beginning of the combined treatment (chemotherapy and extracorporeal treatment) did not differ significantly between the 3 groups of patients (respectively, *p* = 0.427 and *p* = 0.347) (Table 3).

#### 2.1.2. Renal Recovery

As there was no difference in sFLC reduction ratio per session (RRs) between the 3 dialyzers considered, analyses of patients’ renal recovery and survival were performed in the whole sample and not divided among the 3-dialyzer groups. The renal recovery rate was 65% (*n* = 15) and median time to renal recovery was 17 days (QR 10–54). The factors associated to renal recovery were also analyzed. Statistical analysis showed that the only variable associated with dialysis independence, was the number of extracorporeal treatments: median number of dialysis sessions was 20 (min 15–max 30) for patients who did not recover the renal function and 7 (min 3–max 10) for patients who became dialysis independent (*p* < 0.001). Nonetheless, there was no difference in sFLC RRs between these 2 groups (*p* = 0.81). There was a clear difference, close to be statistically significant, in reduction of sFLC at day 12 and day 30 from the beginning of the treatment: patient who recovered the renal function presented a median RR higher than patients who required chronic dialytic treatment: 82% vs. 32% at day 12 (*p* = 0.053), 96.8% vs. 59.1% at day 30 (*p* = 0.093). Likewise, we observed that patients who recovered renal function were younger (64.5 vs. 74 years, *p* = 0.156) and presented higher serum albumin levels (3.6 vs. 3.2, *p* = 0.112) (Table 4).

#### 2.1.3. Patients’ Survival

Statistical analysis showed that higher levels of albumin were significantly associated with a lower 1-year mortality (HR = 0.06, *p* = 0.007) (Figure 1). The other variables considered at the presentation, i.e., sex, age, serum creatinine, proteinuria, sFLC level and β2 microglobulin were unrelated with mortality (Table 5).

In addition, we found that mortality was associated with a higher number of dialysis sessions (HR = 1.03, *p* = 0.035) and that a >50% reduction of sFLC level at 30 days was associated to lower mortality (HR = 0.06, *p* = 0.002) (Figure 2, Table 6). Mortality was also associated with the presence of infections (HR = 4.8, *p* = 0.071) (Table 6). After a follow-up period of 12 months: 6 patients (26%) died (3 women and 3 men) while 17 patients (74%) survived. All non-survivor patients were ISS3 (*p* = 0.050). Among survivors 2 patients (11.8%) required chronic renal replacement therapy while 15 (88.2%) recovered renal function. All non-survivor patients presented an irreversible renal injury and need chronic dialytic treatment.

Since the high rate of renal recovery in survivor group the correlation suggests that renal recovery is the most important factor that associated with patient’s survival (*p* < 0.001) (Figure 3).

## 3. Discussion

In the present study the efficacy and safety of double-filter based extracorporeal removal of sFLC in patients with AKI and newly diagnosed MM was analyzed.

Previously the prospective trial hasn’t shown a beneficial effect of extracorporeal removal of free light chains on survival and renal recovery in patients with cast nephropathy.

Two dialyzers of the same kind were used in series during a 4-h dialysis session. Considering the whole cohort of the patients enrolled, survival improved as an earlier sFLC RR was achieved. The mortality rate was also lower for patients who required a lower number of extracorporeal treatments and presented higher albumin level at baseline. Three kinds of dialyzers with high adsorbitive ability were chosen. As to sFLC and albumin RR, no differences were found between the 3 dialyzers used, no albumin transfusion requirement was assessed but no direct measurement of albumin leakage into the dialysate was carried out. The sFLC RR >50% was assessed as threshold for patients’ survival. These results confirmed the Haynes’s requirements for the survival of patients with AKI from MM: renal recovery, chemotherapy, and higher albumin levels on starting therapy [4]. Similarly, Yadav et al. found that renal recovery and an early high reduction of sFLC were related to patient’s survival. By now it is well known the importance of extracorporeal removal of sFLC thought dialytic treatment in combination to chemotherapy because of prolonged half-life of sFLC in renal impairment and of their tri-compartmental distribution [11,12]. Vice versa chemotherapy alone, not associated with extracorporeal removal of sFLC, could guarantee a dialysis independence rate lower than 20% in patients with AKI and dialysis need [13]. Nonetheless the sFLC distribution volume impairs plasmapheresis efficiency [14], while the use of HCO HD can worsen patient’s survival and required a high rate of albumin transfusions [9]. Nephrologists should also be aware that sFLC are uremic toxins and require adequate removal during dialysis [15].

The need of high adsorptive dialyzers that also avoid albumin leakage leads us to choose both PMMA and PEPA dialyzers (FDX-210GW and FDY-210 GW). Hutchinson and coll. have already analyzed their adsorbing properties in vitro: PMMA determined a sFLC RR of 75%, while PEPA determined a sFLC RR of 55% [2]. To enhance the removal of sFLC, 2 dialyzers for each dialysis session were proposed: the replacement of the 1st dialyzer by a new one takes place after the 2nd dialysis hour to avoid the exhaustion of sFLC adsorption [16]. Sens et al. assessed the efficacy of sFLC removal by using 2 PMMA dialyzers during the same session in patients with AKI from MCN. All the patients required acute dialysis treatment that was carried out by means of daily hemodialysis, 6-h length. This dialysis schedule was maintained for 3 weeks, and dialysis withdrawal was established when sFLC fell <200 mg/L or when the renal recovery was achieved. The study assessed that a significant patients’ survival was achieved if sFLC RR was >50% at day 21, and the renal recovery rate was significant when sFLC RR was >75% at day 12 [10]. As to PEPA dialyzer, our group published the first successful case report of a patient affected by AKI due to MCN who experienced the recovery of renal function by means of chemotherapy combined to extracorporeal removal of sFLC. Two PEPA dialyzers were used in series during the 7 dialysis sessions carried out for sFLC removal [17]. There are no other in vivo experiences of removal of sFLC with PEPA dialyzer.

Previously, some observational prospective trials shown a beneficial effect of extracorporeal removal of free light chains on survival and renal recovery in patients with cast nephropathy. In 2016 Zannetti et al. carried out a monocentric study but they were able to enrol only 21 patients affected by AKI from MM that were treated combining HCO-HD and chemotherapy [18]. Sens et al. in 2017 carried out a multicentric observational study in France but only 17 patients were enrolled and PMMA double filter with daily dialysis was studied [10]. However, these observational trials had some weaknesses: (a) using HCO-HD, Zannetti et al. showed that albumin loss took place and albumin transfusions were required [18]; (b) Sens et carried out a retrospective study where intensive daily dialysis was performed and 2 PMMA dialyzers during 6-h length dialysis sessions were used 6 times per week [10]. Nonetheless, among the observational studies carried out on this field, the number of patients we enrolled is significant. Two randomized controlled trials are available with a huge number of patients enrolled (about 90 patients for each study): namely MYRE and Eulite. Eulite Trial showed a negative outcome of the group who underwent HCO-HD in comparison to the group who underwent high flux HD [9]. The MYRE study showed no benefits on patients’ survival [8].

## 4. Conclusions

The present observational study assessed that in patients with AKI for MM, sFLC removal can be efficiently obtained by means of 2 high flux-adsorbing dialyzers without albumin leakage. The factors that have been found to be significantly associated with higher survival rate were reduction of sFLC, number of extracorporeal session and dialysis independence. Nonetheless the observational design, the lack of a control group and the small number of cases investigated restrict the scope for generalization and prevent us from drawing firm conclusions at present.

## 5. Materials and Methods

A prospective monocentric observational study was carried out to evaluate the efficacy of extracorporeal removal of sFLC combined to chemotherapy in patients with AKI and new-diagnosed MM. The primary endpoint was the evaluation of the 1-year mortality. The secondary endpoints were (a) dialysis independence defined as eGFR over 15 mL/min/1.73 m two weeks after the last dialysis session; (b) the time for renal recovery; (c) the reduction rate of sFLC for each extracorporeal treatment; (d) the reduction rate of sFLC at day 12, day 30 and in the whole period of treatment. Patients with AKI and MM admitted to the Nephrology, Dialysis and Renal Transplant Unit of the Sant’Orsola Hospital were considered in the period November 2014–December 2019. All patients were treated in the Haematology Unit of the same hospital. Inclusion criteria were new-diagnosed MM, AKI defined according to KDIGO guidelines [19], eligibility to chemotherapy with the aim of remission of the disease, eligibility to extracorporeal treatment; sFLC above 1000 mg/L; informed consent.

Bone narrow biopsy was performed in each patient. Renal biopsy was performed according to the clinical status. Renal specimens were analysed for immunofluorescence, light microscopy, and electron microscopy. Extracorporeal treatment was carried out after central venous catheter placement in jugular vein or in femoral vein (Mahurkar^TM^ Medtronic, diameter 12 French, length from 16 to 24 cm).

All dialysis sessions lasted 4 h, circuit anticoagulation was provided thought low molecular weight heparin: sodic enoxaparin (Clexane™, Sanofi-Aventis, Milan Italy) which was administrated at the beginning of dialysis session (2000 UI e.v. in patients with body weight lower than 50 kg, 4000 UI e.v. in patients with body weight between 50 and 90 kg and 6000 UI e.v. in patients with body weight over than 90 kg). Dialytic treatments were carried out every other day at the Nephrology, Dialysis and Renal Transplant Unit of S.Orsola Hospital of Bologna. The dialytic dose was not related to the renal function but to the removal of sFLC. Patients underwent 3 times weekly dialysis session, and 2 dialyzers were used for each session with the exchange of the filter at the second hour with the aim to enhance the removal of sFLC. Blood flow was 300 mL/min; dialysate flow was 500 mL/min. Ultrafiltration rate was regulated according to clinical need of the patient. All the dialyzers used were high flux filters: (a) PMMA filter (Filtryzer BK-F^®^, Toray Co., Tokyo, Japan), neutral surface area of 2.1 m^2^, thickness of 30 μm and ultrafiltration coefficient (Kuf) of 26 mL/h/mmHg, cut off 20,000 daltons [2]; (b) PEPA filter (FDX-210 GW^®^, Nikkiso Co., Tokyo, Japan) surface area of 2.1 m^2^, thickness of 30 μm and Kuf of 63 mL/h/mmHg, cut off 30,000 daltons [20]; (c) PEPA filter (polyester polymer alloy, FDY-210 GW^®^, Nikkiso Co., Tokyo, Japan) surface area of 2.1 m^2^, thickness of 30 μm and Kuf of 64 mL/h/mmHg, cut off 32,600 daltons [20]. PEPA dialyzers were used in ol-HDF. Reinfusion flow was ultrapure online flow lower than 20 L per sessions. This low reinfusion flow was preferred to limit the albumin loss, which has been described with PEPA dialyzers and convective flow above 20 L per sessions [21]. Extracorporeal treatment for removal of sFLC was discontinued when sFLC level was lower than 500 mg/L. Renal recovery was reached if patient was independent from dialytic treatment for almost two weeks from the last dialysis session and had an eGFR above 10 mL/min according to Hutchison et al. [22]. If sFLC normalize, but patients present irreversible renal injury, chronic dialytic treatment was continued, and patients were referred for a stable vascular access according to the need to continue chronic dialytic treatment [23]. In Figure 4 the treatment scheme has been summarized. Blood tests were performed at the beginning of the combined treatment (chemotherapy and extracorporeal treatment) then monthly: (blood cells count, renal function, sFLC, β2 microglobulin, ions, PTH, albumin and plasmatic protein). The dosage of sFLC was carried out at the beginning and at the end of each dialysis session.

Serum level of kappa and lambda light chains was obtained through nephelometry (kit Freelite k/λ, The Binding Site Greey, Birmingham, UK; IMMAGE/IMMAGE 800 Beckman Coulter, Brea, California, USA, Beckman Coulter Beta-2-Microglobulin Kit, Brea California, USA). Range of kappa light chains was 3.3–19.4 mg/L, while range of lambda light chains was 5.7–26.3 mg/L; range of β2 microglobulin was 0.7–2.0 mg/L. Removal of sFLC was estimated through RRs.
RRs = Cpre − Cpost/Cpre(1)

In which Cpre = concentration of the solute on T0, Cpost= concentration of the solute on T1. Since sFLC are comparable to small solutes they uniformly arrange in extracellular water. Their behaviour reflects β2 microglobulin kinetics whom molecular weight is likely to sFLC. All authors that have analysed sFLC removal have considered the monocompartimental model of distribution of sFLC to correct the value measured at the end of dialysis session for the haemoconcentration due to the dehydration of the patient. This model is described through the Bergström and Wehle following formula [24]:Cpost-corr = Cpost/1 + (ΔBW/0.2 × BW post)(2)

C post-corr is the correct concentration of the solute, Cpost is the concentration measured at the end of a single dialysis session, ΔBW (Delta body weight) is the intradialytic weight loss, BW post is the weight of the patient at the end of the dialysis after water removal, 0.2 correspond to the 20 % of the body weight which is the distribution volume of β2 microglobulin. The stop dialysate flow method, that also involves the slow blood flow before blood drawing, was used to avoid access recirculation [25].

## 6. Statistical Analysis

The statistical analysis was performed using Stata statistical software, version 15.1 (StataCorp. 2017. *Stata Statistical Software: Release 15*. College Station, TX: StataCorp LLC). The normality of the distribution of continuous variables was tested by means of Shapiro-Wilk’s test. All continuous variables had a non-normal distribution; therefore, they were summarized using median and interquartile range [IQR]. Clinical and demographic characteristics were compared among groups using the non-parametric Kruskal-Wallis test (comparison between the three types of dialyzer) and Mann-Whitney test (comparison between renal recovery groups). 1-year survival was estimated using Kaplan-Meier method and Cox regression analysis. The level of statistical significance was set at *p* < 0.05.

## Figures and Tables

**Figure 1 toxins-14-00391-f001:**
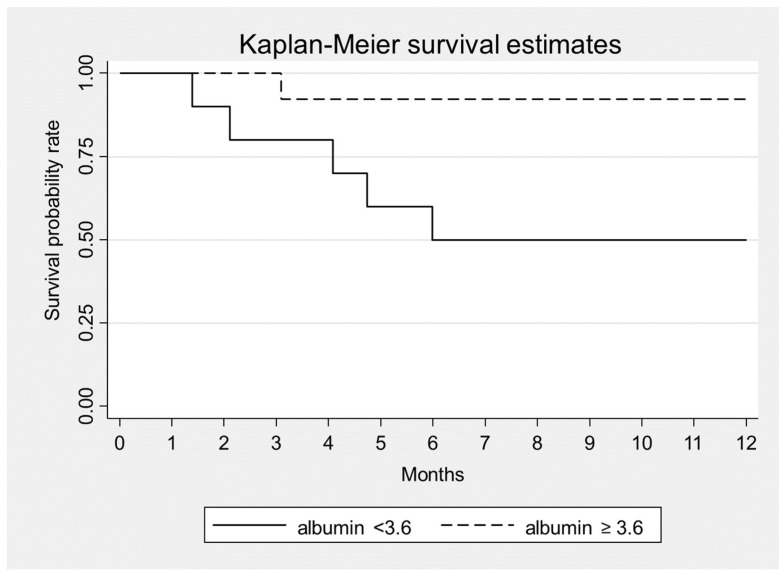
Kaplan–Meier survival estimates according to serum albumin at baseline, *p* = 0.007.

**Figure 2 toxins-14-00391-f002:**
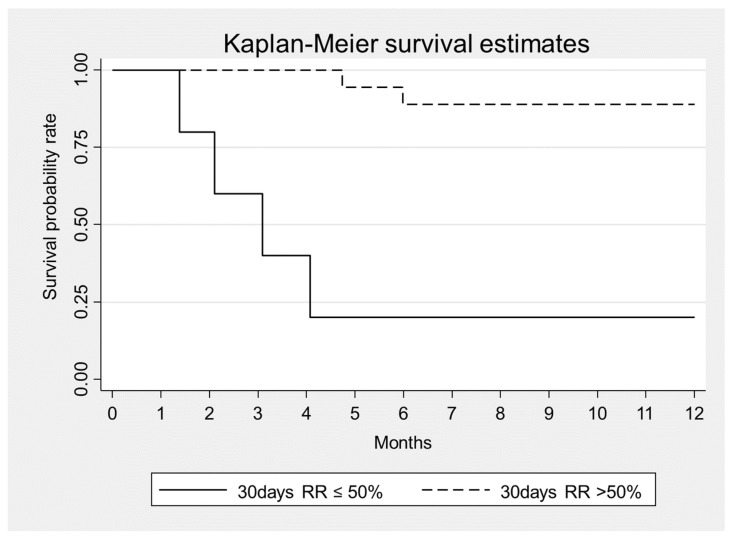
Kaplan–Meier survival estimates according to sFLC removal at Day 30 from the beginning of combined treatment, *p* = 0.002.

**Figure 3 toxins-14-00391-f003:**
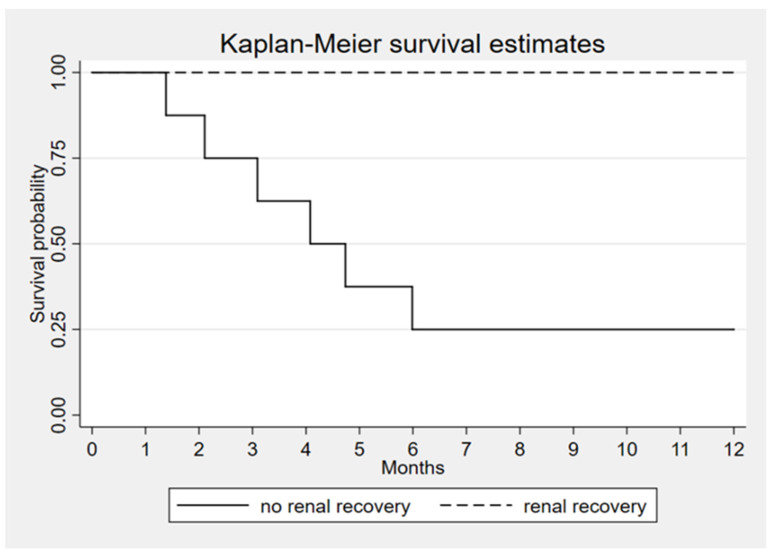
Kaplan–Meier survival estimates according to renal recovery, *p* < 0.001.

**Figure 4 toxins-14-00391-f004:**
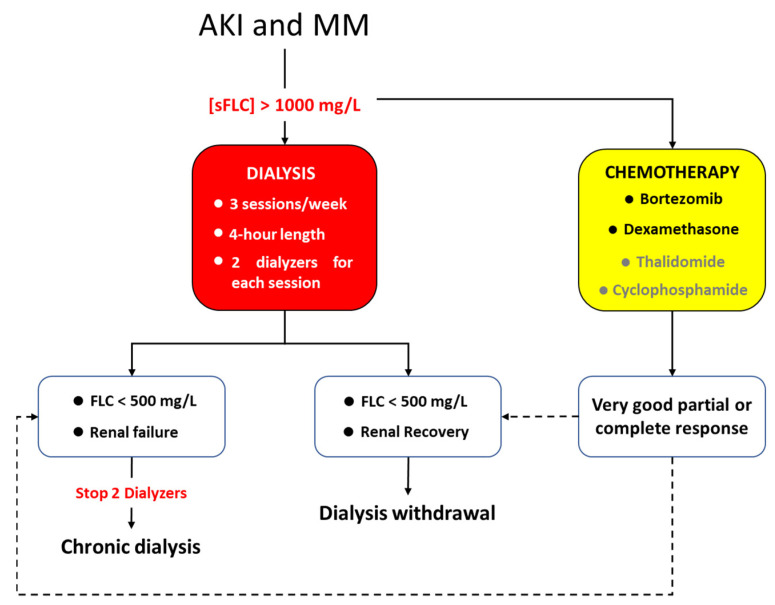
Scheme with all the steps of the study.

**Table 1 toxins-14-00391-t001:** Patients’ characteristics at baseline.

Patients (*n*)	23
Median age (range) (years)	67 (41–85)
Males/Females (*n*)	16/7
Diabetes (%)	6 (26.08)
Hypertension (%)	11 (42.3)
Previous CKD (%)	8 (23.78)
ISS3 (%)	15 (65.2)
Sub-type of monoclonal antibody -IgG kappa, patients (n) -IgG lambda, patients (*n*) -IgA kappa, patients (*n*) -Bence Jones lambda, patients (*n*) -FLC kappa, patients (*n*)	13 1 5 1 3
Light chain involved (%) -kappa, patients (%) -lambda, patients (%) -kappa FLCs, median (range) (mg/L) -lambda FLCs, median (range) (mg/L)	16 (69.6) 7 (30.4) 3475.6 (1000.1–15,768) 11,279 (4698.1–22,286)
Uric acid, median (range) (mg/dL)	7.2 (3.4–16.9)
eGFR (CKD-EPI), median (range) (mL/min)	13.72 (2.8–57)
Creatinine, median (range) (mg/dL)	6.2 (1.1–15)
Hb, median (range) (g/dL)	9.4 (7.3–13)
Proteinuria, median (range) (mg/dL)	1200 (30–25,860)
Serum albumin, median (range) (mg/dL)	3.6 (2.3–4.7)
Calcium, median (range) (mg/dL)	9.8 (7–16)
Bone involvement (%)	20/23 (86.9)
PTH, median (range) (ng/mL)	26 (8–217)
Alkaline phosphatase, median (range) (U/L)	48 (43–1062)
Renal Histology -No biopsy (n) -MCN, (*n*) -MIDD, (*n*) -LCDD, (*n*)	5 14 3 1

Free Light Chain (FLC); Haemoglobin (Hb); Estimated glomerular filtration rate (eGFR); Chronic Kidney Disease Epidemiology Collaboration (CKD-EPI); parathyroid hormone (PTH); Myeloma Cast Nephropathy (MCN); Monoclonal immunoglobulin deposition disease (MIDD); Light chain deposition disease (LCDD); International Staging System 3 (ISS3).

**Table 2 toxins-14-00391-t002:** Patients’ characteristics for the 3 dialyzers used at baseline.

	PMMA (*n* = 7)	FDX (*n* = 7)	FDY (*n* = 9)	*p*-Value
Age (years), median [IQR]	67.5 (66–74)	75.7 (58–85)	64 (58–69)	0.254
Male Sex, *n* (%)	4 (57.1)	5 (71.4)	7 (77.8)	0.849
Creatinine (mg/dL), median [IQR]	6.6 (5.4–12)	6.0 (3.1–7)	5.5 (2.1–8.3)	0.306
eGFR (mL/min), median [IQR]	7.5 (3.7–8.4)	7.8 (6.9–12.3)	10 (6.3–32.7)	0.374
Proteinuria (mg/day), median [IQR]	570 (30–2550)	452 (8–5000)	2800 (360–11,984)	0.158
sFLC Kappa, *n*	6	4	6	-
sFLC Kappa at baseline (mg/L), median [IQR]	3662 (2160–15,100)	5752 (3933–8681)	1288 (1004–3971)	0.113
sFLC Lambda, *n*	1	3	3	-
sFLC Lambda at baseline (mg/L), median [IQR]	645	6438 (4698–13,336)	11,984 (11,279–22,286)	0.208
Beta 2 microglobulin (mg/L), median [IQR]	10 (5.2–11.2)	29.4 (12.5–35)	10.7 (6–22)	0.108
Albumin (g/dL), median [IQR]	2.8 (2.7–3.6)	3.7 (3.3–3.8)	3.6 (3.6–3.8)	0.103

Serum free light chains (sFLC) estimated glomerular filtration rate (eGFR). Interquartile range (IQR), polymethyl methacrylate (PMMA), serum Free Light Chain (sFLC).

**Table 3 toxins-14-00391-t003:** sFLC and albumin removal with the 3 dialyzers considered.

	PMMA (n = 7)	FDX (n = 7)	FDY (n = 9)	*p*-Value
sFLC RRs (%), median [IQR]	50.4 (39.2–53.0)	51.2 (23.8–53.2)	49.7 (40.3–55.6)	0.698
RR day 12 (%), median [IQR]	51.4 (29.5–92.6)	75.6 (33.3–80.3)	84.0 (58.2–89.7)	0.427
Patients with RR ≥ 50% at day 12, *n* (%)	3 (50)	5 (71.4)	7 (77.8)	0.538
RR day 30 (%), median [IQR]	90.8 (−43.9–98.4)	96.8 (57.6–98.9)	94.9 (80.7–99.7)	0.347
Patients with RR ≥ 50% at day 30, *n* (%)	4 (57.1)	6 (85.7)	8 (88.9)	0.445
Albumin (RR, %), median [IQR]	15 (5–24)	10 (1–25)	9 (2–27)	0.531

Serum free light chains (sFLC), reduction ratio (RR), interquartile range (IQR). serum Free Light Chain (sFLC).

**Table 4 toxins-14-00391-t004:** Renal recovery.

	NO (*n* = 8)	YES (*n* = 15)	*p*-Value
Extracorporeal treatments (*n*), median [IQR]	20 (15–30)	7 (3–10)	0.001
sFLC RRs (%), median [IQR]	50.8 (43.2–52.2)	49.7 (36.8–55.2)	0.821
RR day 12 (%), median [IQR]	32.4 (24.9–75.6)	81.8 (54.1–90.5)	0.053
RR day 30 (%), median [IQR]	59.1 (−30.4–96.7)	96.8 (80.9–99.0)	0.093
Age (years), median [IQR]	74 (67–79)	64.5 (58–69)	0.156
Male Sex, *n* (%)	5 (62.5)	11 (73.3)	0.657
Diabetes, *n* (%)	3 (37.5)	3 (20.0)	0.621
Hypertension, *n* (%)	3 (37.5)	8 (53.3)	0.667
Pre-existing CKD, *n* (%)	4 (50.0)	4 (26.7)	0.371
ISS3, *n* (%)	7 (87.5)	8 (53.3)	0.307
Creatinine (mg/dL), median [IQR]	6.6 (5.8–9.5)	5.5 (2.7–8.3)	0.175
eGFR (mL/min), median [IQR]	7.3 (5.3–8.3)	9.7 (6.3–23.3)	0.220
Proteinuria (mg/day), median [IQR]	100 (40–2600)	452 (8–4500)	0.583
sFLC Kappa, *n*	6	10	-
sFLC Kappa at baseline (mg/L), median [IQR]	7722 (2980–15,100)	2392 (1160–5242)	0.065
sFLC Lambda, *n*	2	5	-
sFLC Lambda at baseline (mg/L), median [IQR]	6990 (645–13,336)	11,279 (6438–11,984)	0.699
Beta 2 microglobulin (mg/L), median [IQR]	11.1 (7.6–34.8)	12.5 (6.8–29.4)	0.651
Albumin (g/dL), median [IQR]	3.2 (2.7–3.7)	3.6 (3.4–3.8)	0.112

Serum free light chains (sFLC) estimated glomerular filtration rate (eGFR), reduction ratio (RR); interquartile range (IQR); International Staging System 3 (ISS3).

**Table 5 toxins-14-00391-t005:** Univariate analysis of baseline patients’ characteristics and 1-year mortality. Hazard Ratio (HR) and 95% CI.

	HR	95% CI	*p*
Sex	0.38	0.08–1.88	0.236
Age (years)	1.04	0.96–1.12	0.321
Diabetes	1.25	0.23–6.85	0.795
Hypertension	0.52	0.10–2.85	0.452
Pre-existing CKD	2.09	0.42–10.40	0.367
Serum Creatinine (mg/dL)	1.15	0.93–1.42	0.207
Proteinuria (>1000 mg/day)	0.98	0.95–1.02	0.325
sFLC (mg/L)	1.00	0.98–1.01	0.702
Beta 2 microglobulin (mg/L)	0.99	0.93–1.06	0.754
Serum Albumin (g/dL)	0.06	0.01–0.47	0.007

**Table 6 toxins-14-00391-t006:** Univariate analysis of treatment period characteristics and 1-year mortality.

	HR	95% CI	*p*
Extracorporeal treatments (*n*)	1.03	1.00–1.06	0.035
sFLC RRs (%)	0.69	0.01–83.7	0.879
RR albumin (%)	0.81	0.1–1.9	0.821
RR day 12(%)	0.95	0.91–0.99	0.011
RR > 50% day 12	0.08	0.01–0.76	0.028
RR day 30 (%)	0.97	0.95–0.99	0.004
RR > 50% day 30	0.06	0.01–0.34	0.002
Infections	4.8	0.9–26.5	0.071

Reduction ratio per session (RR).

## Data Availability

The datasets used and analysed during the current study are available from the corresponding author on reasonable request.

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
