# Peer review of "Early Light Chains Removal and Albumin Levels with a Double Filter-Based Extracorporeal Treatment for Acute Myeloma Kidney"

_toxins, 2022, doi:10.3390/toxins14060391_

Round 1

Reviewer 1 Report

The authors studied hemodialysis status in myeloma patients with renal insufficiency and concluded that albumin level was associated with patients' outcome.

  1. The study had bias on the retrospective designed and small number study which hamper a clear conclusion.
  2. The authors had 3 type of dialysis method, however, no difference effects by these three methods, which not meet their primary goal.
  3. The dialysis criteria is not the same, some are in need for dialysis, some are not, which will impact on the conclusion. Not to say, the small numbered of patients in these three groups could not comes out with good results.
  4. The authors suggested that the reduction of sFLC at different time point associated with outcomes, however, the reduction of sFLC comes not only from dialysis, but also could be from treatment response. Therefore, treatment response will associate with outcome. The author should proof the reduction were from the dialysis.
  5. In the study, the survival was associated with albumin level, which is already known and already been cooperated into staging system. Therefore, the conclusion add few insight into the field, especially with small numbered patients. 

Reviewer 2 Report

In this article, the authors deal about Early Light Chains Removal and Albumin Levels with a Double Filter-Based Extracorporeal Treatment for Acute Myeloma Kidney.

The topic is of interest and the manuscript is well written. I recommended that this article is acceptable for publication after revision.

Comments and suggestions:

Introduction section: Add more data about MM and AKI. Also, the acronyms of all clinical trials mentioned should be explained, for example, European Trial of Free Light Chain Removal by Extended Haemodialysis in Cast Nephropathy (EuLITE), and so on.

A scheme with all the steps of this study is necessary to understand it better.

For all tables: all abbreviations included in them must be explained.

What perspectives for human health does this MS have?

Consider revision accordingly.

Reviewer 3 Report

The submitted publication is of interest and contains some new information about the use of Double Filter-Based Extracorporeal Treatment for Acute Myeloma Kidney. The overall design of the study and the results presented do not raise any particular doubts about the quality. The remark that can be made is, in my opinion, the need for a more detailed discussion with a discussion of the results of the implementation of alternative options for reducing the level of Early Light Chains and albumin.

Round 2

Reviewer 1 Report

The authors response and modified the text accordingly to reviewers' comments. I think it is suitable to be published in the journal.